# Chemotherapy-Induced Liver Injury in Patients with Colorectal Liver Metastases: Findings from MR Imaging

**DOI:** 10.3390/diagnostics12040867

**Published:** 2022-03-31

**Authors:** Francescamaria Donati, Dania Cioni, Salvatore Guarino, Maria Letizia Mazzeo, Emanuele Neri, Piero Boraschi

**Affiliations:** 1Department of Diagnostic, Interventional Radiology and Nuclear Medicine, Pisa University Hospital, Via Paradisa 2, 56124 Pisa, Italy; p.boraschi@gmail.com; 2Academic Radiology, Department of Translational Research, University of Pisa, Via Roma 67, 56126 Pisa, Italy; cioni.dania@yahoo.com (D.C.); marialetiziamazzeo@gmail.com (M.L.M.); emanueleneri1@gmail.com (E.N.); 3Department of Radiology, Monaldi Hospital, AORN dei Colli, Str. Vicinale Reggente 66/82, 80131 Naples, Italy; sag1981@libero.it

**Keywords:** chemotherapy-associated steatohepatitis, hepatic damage, peliosis, nodular hyperplasia, sinusoidal obstruction syndrome, magnetic resonance imaging

## Abstract

Chemotherapy-induced liver injury has been found to be quite common in cancer patients undergoing chemotherapy. Being aware of chemotherapy-induced hepatotoxicity is important for avoiding errors in detecting liver metastases and for defining the most appropriate clinical management strategy. MRI imaging has proven to be a useful troubleshooting tool that helps overcome false negatives in tumor response imaging after chemotherapy due to liver parenchyma changes. The purpose of this review is, therefore, to describe the characteristics of magnetic resonance imaging of the broad spectrum of liver damage induced by systemic chemotherapeutic agents in order to avoid misdiagnoses of liver metastases and disease progression and to define the most appropriate clinical management strategy.

## 1. Introduction

Nowadays, chemotherapy-induced liver injuries are common findings from clinical practice in the imaging of oncology patients undergoing neo-adjuvant chemotherapy. Rubbia-Brandt et al. [1] were the first to systematically analyze the morphological effects of neo-adjuvant systemic chemotherapy on non-tumor-bearing hepatic parenchyma, reporting a high prevalence of sinusoidal obstruction syndrome (SOS) in patients with metastatic colorectal cancer treated with various protocols, including oxaliplatin. Two main forms of chemotherapy-induced liver injuries have been described: hepatic steatosis and SOS^2^. Furthermore, chemotherapy-induced liver injuries constitute a significant issue in patients undergoing liver resection for hepatic metastases as they affect postoperative morbidity and mortality to varying degrees [2]. The aim of this review is to describe the features of magnetic resonance imaging (MRI) of the broad spectrum of liver injuries induced by systemic chemotherapeutic agents in order to avoid misdiagnoses of hepatic metastasis and disease progression and to define the most appropriate clinical management strategy.

## 2. Hepatic Steatosis, Steatohepatitis, and Cirrhosis

Chemotherapeutic agents may lead to non-alcoholic fatty liver disease, a wide spectrum of diseases ranging from simple hepatic steatosis to a more severe steatohepatitis [3,4]. Interestingly, 5-fluoroacil (5-FU) and irinotecan, chemotherapeutic agents administered to patients with colorectal cancer with metastases, have been reported to be closely related to chemotherapy-induced fatty liver disease [5]. Fatty liver disease has also been reported to be related to other chemotherapeutic agents, including platinum derivatives, taxanes, irinotecan, cetuximab, L-asparaginase, dactinomycin, mitomycin C, bleomycin sulfate, and methotrexate [6]. In the case of oxaliplatin, no significant association with hepatic steatosis or steatohepatitis has been found [7]. However, the frequency of this disease has not yet been determined [6]. Hepatic steatosis is characterized by excessive triglyceride accumulation and the deposition of lipid vesicles within the cytoplasm of hepatocytes [5,8]; either microvesicular or macrovesicular steatosis can be observed in chemotherapy-induced hepatic steatosis [9]. The overlap among lobular inflammation, and the ballooning degeneration of hepatocytes and eventually of fibrosis is suggestive of steatohepatitis [4]. Steatohepatitis is associated with a high risk of developing cirrhosis and its complications, such as liver failure and hepatocellular carcinoma. Concerning pathogenesis, chemotherapeutic agents promote oxidative stress and mitochondrial accumulation of large amounts of reactive oxygen species not only in cancer cells but also in normal hepatocytes, inducing the deposition of lipid vesicles, which can be diffuse or focal [4,5,6]. Notably, the focal deposition of lipid vesicles can be the cause of a diagnostic challenge because it can mimic liver metastasis [5].

Identifying and monitoring chemotherapy-induced fatty liver disease are important for several reasons. First, chemotherapeutic agents should continue to be used even after the detection of fatty liver disease and monitoring the signs of progressive liver damage is important [9]. Second, hepatic steatosis, especially, steatohepatitis, has been reported to be a risk factor for increased perioperative morbidity and mortality in patients undergoing major hepatic resection for colorectal cancer liver metastases [10]. Furthermore, radiologists should be prudent in avoiding administering 5-FU and irinotecan in patients with severe steatosis identified on imaging [11]. Due to the important clinical impact of this disease, diagnosis and quantification of hepatic fatty infiltration and inflammation are fundamental after chemotherapy [4].

Liver biopsy is the gold standard for distinguishing between steatosis and steatohepatitis and for characterizing the stages of lipid vesicles in hepatocytes, lobular inflammation, ballooning degeneration of hepatocytes, and hepatic fibrosis [3,6]. However, some potential drawbacks of liver biopsy exist and evaluating patients with liver biopsy in every follow-up period is not feasible [9].

Hence, imaging modalities are commonly used for this purpose [12,13,14]. Particularly, MRI represents a good and non-invasive alternative to liver biopsy for the diagnosis; quantification; and above all, monitoring of fatty liver disease [12].

Primarily, MRI allows for a qualitative diagnosis of steatosis, which shows signal loss on opposite-phase T1-weighted images compared with in-phase images [6]. In contrast, no signal drop is seen in the opposite-phase images of metastasis [6] (Figure 1). Furthermore, MRI provide accurate non-invasive quantification of intrahepatic fat using several advanced techniques [9]. The principle of MRI to quantify fat depends on the chemical-shift effect, which can be defined as the difference in resonance frequencies between hydrogen protons bound to triglycerides and water. This difference can be seen directly on the spectra in magnetic resonance spectroscopy (MRS) or can be calculated as the proton density fat fraction (PDFF) using different MRI techniques [9]. MRS is a technique capable of showing the molecular composition of certain tissue as resonance peaks at different locations on the spectra. On the MRS spectra of the liver, two main peaks can be seen: water positioned at 4.7 ppm and fat positioned at 1.3 ppm [9]. The signal intensities of these peaks can be quantified by spectral tracing of the peaks, and fat content can be calculated by giving the ratio of signal intensities of fat peaks to the sum of the fat and water peaks. Single-voxel spectroscopy is a common tool used for the quantification of fat in the liver. In single-voxel spectroscopy, data are collected from a single voxel, which is placed on the interested liver parenchyma, avoiding vessels, bile ducts, and surrounding adipose tissue [9]. High intra-individual reproducibility was also reported for MRS. Therefore, MRS was accepted as a reference imaging method for the assessment of hepatic steatosis. However, MRS also has some drawbacks: it demonstrates a fat fraction of a limited portion of the liver; is conditioned by the differences in fat distribution among different regions of the liver; is not available on all clinical scanners; and requires postprocessing software and specific analyses, which limits its usage in daily routine [9]. These drawbacks accelerated the development of other MRI techniques for the quantification of hepatic steatosis [13]. In this regard, another advanced MRI technique for intrahepatic fat quantification is to calculate the proton density fat fraction (PDFF) of the liver by separating the signals from water and fat [9] (Figure 2). The PDFF is defined as the ratio between the density of hydrogen protons from liver fat and the total hydrogen proton density from all mobile proton species [5]. Its advantages over MRS include simplicity and the ability to quantify steatosis in the whole liver within a short time (<20 s) [5,13]. Several studies have suggested a high diagnostic accuracy for PDFF [15,16]. Particularly, PDFF was reported to have an accuracy of 100% for detecting an abnormal quantity of hepatic steatosis when the value was higher than 5.7% [16]. Other studies showed a close correlation between histology-determined steatosis and MRI-PDFF-determined steatosis, with high accuracy rates [17]. Moreover, this technique can also be used for the quantification of longitudinal changes in hepatic fat content in patients with fatty liver disease [18]. Studies comparing MRI-PDFF with MRS-determined hepatic steatosis reported an excellent correlation [18]. The advantages of MRI-PDFF over MRS also include simplicity and the ability to quantify steatosis in the whole liver within a short time. However, MRI-PDFF is not feasible in many patients as it is expensive and not widely available [3]. While the abovementioned MRI advanced techniques provide accurate detection and quantification of intrahepatic fat, hepatic inflammation cannot yet be precisely and non-invasively quantified from imaging [4]. In the last few years, LiverMultiScan™ (LMS, Perspectum Diagnostics, Oxford, UK), a multiparametric MRI-based method, has emerged as a promising diagnostic tool for diagnosing, quantifying, stratifying, and monitoring steatohepatitis [19]. Moreover, MR elastography is used in many tertiary centers as a non-invasive technique for the detection and staging of liver fibrosis and for the differentiation of isolated fatty liver disease from steatohepatitis with or without fibrosis. However, extensive validation of these promising recent MRI techniques is needed [4].

## 3. Sinusoidal Obstruction Syndrome (SOS)

Sinusoidal obstruction syndrome (SOS), previously known as veno-occlusive disease, includes a continuum of hepatic histological lesions ranging from simple sinusoidal dilation to nodular regenerative hyperplasia (NRH) and is the most common hepatic complication after chemotherapy [1,20,21,22]. In particular, oxaliplatin, which is an essential part of chemotherapeutic regimens (FOLFOX and XELOX) for colorectal cancer has been reported as an important contributor to SOS development, as initially described by Rubbia-Brandt [1,20,23]. The incidence of oxaliplatin-induced SOS verified by histopathologic examination in the literature has been described as variable between 18% and 59% of patients [1,23,24].

Several studies have analyzed the clinical consequences of SOS among patients undergoing liver resection for colorectal cancer liver metastasis, showing that chemotherapy-induced SOS is associated with a higher rate of postoperative complications and longer hospital stay. Indeed, chemotherapy-induced SOS increases the risk of perioperative bleeding, which requires major hepatectomies and multiple red blood cell transfusions, and postoperative liver failure, particularly for high-grade SOS [25].

However, bevacizumab has been reported to have a protective effect against oxaliplatin-induced hepatopathy, reducing the onset and severity of SOS [20,22,26,27,28]. This protective effect, specific to anti-VEGF agents, was partially explained by Deleve et al. [29] and Iguchi et al. [30], who found that SOS severity was correlated with circulating VEGF levels.

Other causes of SOS include hematopoietic stem cell transplantation (HSCT), ingestion of toxic alkaloids, high dose radiation therapy, and liver transplantation [21,22,31,32,33].

Concerning pathophysiology, SOS is caused by toxic injury to the endothelium of the hepatic sinusoids [4,21,23]. In the acute phase, the chemotherapeutic drugs cause loss of sinusoidal wall integrity with extravasation of erythrocytes within the space of Disse and endothelial cell exfoliation with downstream embolization, inducing obstruction of hepatic sinusoids and centrilobular hepatic veins, sinusoidal dilatation, and hepatocellular congestion in centrilobular areas, which can lead to centrilobular hepatocellular necrosis [4,21,23,31]. In the subacute phase occurring days or weeks after injury, extravasation of erythrocytes and accumulation of hemosiderin within the space of Disse favor the proliferation and activation of hepatic stellate cells and subendothelial fibroblasts, resulting in the deposition of the extracellular matrix in perisinusoidal spaces and in the centrilobular vein, with progressive obliteration of the sinusoids and venules [4,21,23,31]. Finally, when SOS persists for weeks, months, or even years, the sinusoidal obstruction and consequent reduction in blood flow in the sinusoids lead to portal hypertension, liver dysfunction, and destruction of the hepatic parenchyma with nodular regeneration [4,21,23,31].

Thus, a radiologist needs to be able to recognize its typical features during cancer monitoring before surgery and to identify the best timing for liver resection or for further chemotherapy. Liver biopsy is the gold standard method for confirming a diagnosis of SOS, but this approach is invasive and carries a high risk of infection, rendering its routine use impractical [21,22].

Patchy heterogeneity of hepatic parenchyma on contrast-enhanced MR images, with a scattering of micronodular or large hypointense areas, is the most important imaging feature of SOS and usually leads a radiologist to consider this condition [21,22,34,35,36,37,38].

Although oxaliplatin-induced SOS may have a diffuse distribution, it shows a predominant distribution at the periphery of the right hepatic lobe. This predominance can be explained by the features of the portal flow carrying the drug [12] and the prevalence of sinusoidal injuries in subcapsular areas [20,22,34,35].

Since the initial lesion of SOS is located in hepatic sinusoid, the first structures to be involved are centrilobular and sublobular veins with blood stagnation, hepatocellular edema, and necrosis, inducing defects of liver enhancement and patchy parenchymal heterogeneity [20,22].

Zhou et al. [33] found a significant association between the extent of patchy parenchymal heterogeneity on enhanced MRI images and the clinical severity of SOS. Furthermore, according to Han et al. [23], the severity of hepatic parenchymal heterogeneity typical of SOS affects the tumor response of hepatic metastases of colorectal cancer to oxaliplatin-based chemotherapy. The results of their study showed that the more severe the hepatic parenchymal heterogeneity, the worse the expected tumor response of colorectal cancer liver metastasis to oxaliplatin-based chemotherapy.

In contrast, Staal et al. questioned the routine use of MRI for SOS evaluation. In fact, while severe cases are easily recognized, mild cases still require a great deal of experience from the radiologist to be diagnosed [39].

However, MRI with gadoxetic acid, a liver-specific contrast agent with combined perfusion and hepatobiliary properties, is a very useful tool for assessing the function of hepatocytes and for detecting SOS in its relatively early stage [32].

Shin et al. [36] reported that the presence of hepatic reticular hypointensity in the hepatobiliary (HB) phase of gadoxetic acid-enhanced MRI was highly specific for the diagnosis of SOS (Figure 3).

Yoneda et al. [32] found that reticular hypointensity of SOS in the HB phase was due to damage to centrilobular hepatocytes, expressing organic anion transporting polypeptide (OATP) 1B3 in normal liver parenchyma, OATP1B3 is the main uptake transporter of gadoxetic acid into hepatocytes and is almost exclusively expressed in centrilobular hepatocytes, representing the primary site of damage in SOS.

Due to its unique ability to allow for tissue differentiation based on architectural changes and cell density, diffusion-weighted MR-imaging (DW-MRI) is known to aid in the assessment of metastatic liver [37,38]. However, specific characteristics are not recognized in the SOS assessment; irregular parenchymal heterogeneity is characterized by the isointensity of the signal on DW-MRI, with a high *b*-value and a low value of the apparent diffusion coefficient (ADC), which is significantly higher than that of metastases [6].

Gradual hepatic functional recovery after interruption of chemotherapy has also been reported in the literature [22,32].

### Focal Chemotherapy-Induced Hepatopathy

A radiologist should be aware that sometimes oxaliplatin-induced SOS may manifest as new focal liver lesion, also known as focal chemotherapy-induced hepatopathy [38], mimicking liver metastasis and constituting a new challenge [38,40]. This is a comprehensive concept that includes several pathologic conditions such as pseudotumor, peliosis, and nodular regenerative hyperplasia [6]. According to Han et al. [38], focal chemotherapy-induced hepatopathy on gadoxetic acid-enhanced MRI shows no rim-enhancement during the arterial and portal phases, intermingled hypointensity and ill-defined margins during the HB phase, and a lack of diffusion restriction on DW-MRI [4,38].

Sometimes in the setting of SOS, a hepatic pseudotumor, characterized pathologically by sinusoidal dilatation and congestion of the hepatocytes with inflammatory cellular infiltration and areas of fibrosis, may arise [41]. On MRI, a pseudotumor of SOS must be suspected in the case of an irregular lesion with peritumoral enhancement and central low attenuation or signal intensity [21]. As a pseudotumor of SOS may simulate focal nodular hyperplasia, atypical HCC, or liver metastasis, DW-MRI is recommended for improving differential diagnostic efficiency between a pseudotumor of SOS and malignancy [38,41] (Figure 4). In fact, while the metastases are hyperintense in DW-MRI images at high *b*-values and have much lower ADC values than normal hepatic parenchyma, a hepatic pseudotumor appears isointense at high *b*-values and has ADC values higher than the metastases but lower to those of benign liver lesions. This can be explained by the increased physical barrier and decreased perfusion, as suggested by Zhang et al. [12] in patients with diffuse liver injury from chemotherapy.

Another uncommon focal liver lesion that may arise in the spectrum of oxaliplatin-induced sinusoidal damage is hepatic peliosis. This condition is pathologically characterized by multiple or occasionally solitary mottled blood-filled cyst-like spaces in the liver with associated sinusoidal dilatation [4]. MRI findings of hepatic peliosis depend on the stages of the blood component in these lesions. In T1-weighted images after contrast material injection, peliotic lesions usually show typical centrifugal enhancement (from the center to the periphery of the lesion); in DW-MRI, peliosis is characterized by hyperintense areas with similar diffusion restrictions to hemangioma, while in the MRI HB phase, these lesions appear as hypointense areas because the blood-filled cavities lack functional hepatocytes (Figure 5). However, some lesions may show central enhancement, which is suggestive of a spared, normal hepatocyte area [42]. Liver biopsy is indicated in case of doubt to differentiate liver metastases from peliosis.

The end and most significant stage of the continuum of histological lesions typical of chemotherapy-induced SOS is represented by nodular hyperplasia.

Nodular hyperplasia is characterized by non-neoplastic regenerative nodules that are not surrounded by fibrous septa [4].

Post-chemotherapy regenerative nodules may develop in response to hyperperfusion and are made up by hyperplastic hepatocytes without atypia [4]. In particular, post-chemotherapy regenerative nodules include monoacinar regenerative nodules, also known as nodular regenerative hyperplasia (NRH), and multiacinar regenerative nodules, also known as focal nodular hyperplasia (FNH)-like lesions [4].

NRH is characterized by regenerative nodules usually > [3] mm in size. Concerning pathophysiology, NRH is believed to be secondary to SOS-induced changes in intrahepatic blood flow, leading to atrophic hypoperfused areas intermingled with hyperperfused regenerative areas [1,2,20]. NRH is generally considered hepatocytic hyperplasia with little pathological significance. In fact, NRH is known to regress up to nine months after the last cycle of oxaliplatin-based chemotherapy [27].

However, in its more widespread and advanced forms, NRH nodules may lead to compression and atrophy of the surrounding liver parenchyma, portal hypertension, and increased postoperative morbidity [4].

A prevalence of NRH of approximately 15–20% has been reported after oxaliplatin-based chemotherapy [2,4,20,27,43]. On the contrary, the association of bevacizumab has a protective effect and reduces the prevalence of NRH [1,2,20,27]. However, no correlation was demonstrated between NRH and the number of chemotherapy cycles [2,4,43].

NRH appears hypointense in the T1-weighted images and hyperintense in the T2-weighted images; the gadoxetic acid-enhanced MRI shows the lesions as slightly enhanced areas in the vascular phase and as hypointense areas in the HB phase; and the absence of diffusion restrictions in DW-MRI excludes the possibility of liver metastases [6,44,45,46] (Figure 6).

FNH-like lesions are also considered a late manifestation of SOS; in a multi-institutional study, Furlan et al. [47] reported a case series of 14 patients with FNH-like lesions diagnosed from liver MRI and occurring de novo after oxaliplatin-based chemotherapy.

The exact pathogenesis leading to the occurrence of FNH-like lesions after oxaliplatin-based chemotherapy remains unknown [47]. However, the changes in hepatic perfusion leading to NRH are also thought to be responsible for the formation of FNH-like lesions [47].

Oxaliplatin-induced FNH-like lesions are usually abundant and up to 42% of them may increase in size when viewed during follow-up imaging [47].

The mean interval between the completion of oxaliplatin-based chemotherapy and the identification of new hepatic FNH-like lesions was 47.6 months (median, 42.5 months; range, 10–120 months) [4,47]. On the contrary, the interval for the occurrence of hepatic metastasis from colorectal cancer is much shorter [47]. Walter et al. [48] reported that no hepatic metastasis from colorectal cancer was detected at CT later than 3.5 years after surgery.

Due to the development of multiple de novo lesions after chemotherapy and the potential increase in size, a radiologist should know the association between oxaliplatin and benign hepatic FNH-like lesions in order to avoid a misdiagnosis of liver metastasis [4,47].

In MR imaging, FNH-like lesions usually show strong enhancement in the arterial phase and isoattenuation to the surrounding liver parenchyma during the portal venous and delayed phases [4]. A central scar is present in fewer than 50% of cases [47]. The specificity for the diagnosis of these benign lesions with non-invasive imaging has improved with the use of hepatobiliary MR contrast agents [4,48,49,50]. After the injection of gadoxetic acid and gadobenate dimeglumine, FNH-like lesions are usually iso- to hyperintense during the MRI HB phase because of a similar or stronger OATP1B3 expression than in the background liver [49]. In addition, a ring of enhancement may be present in up to 50% of cases and is characterized by doughnut-like enhancement (hyperintense periphery and a hypointense center) [47] (Figure 3). Unlike metastases, which appear as hypointense lesions during the HB phase of MRI, the isointensity or hyperintensity of FNH-like lesions is highly specific [34].

As reported by Vigano et al. [34], it is important to remember that SOS and regenerative hyperplasia regress only after nine months without chemotherapy, thus affecting the timing of any liver resections for hepatic metastasis.

## 4. Conclusions

Chemotherapy-associated liver injuries are common findings during routine surveillance from oncologic imaging of patients undergoing neo-adjuvant chemotherapy, and they include hepatic steatosis; steatohepatitis; and the continuum of histological lesions typical of SOS, ranging from simple sinusoidal dilation to nodular regenerative hyperplasia.

These lesions constitute a significant issue in patients undergoing chemotherapy because they can both mimic liver metastases on tomographic imaging and affect postoperative morbidity and mortality to varying degrees after liver resection for hepatic metastasis.

A thorough understanding of the MRI features of chemotherapy-induced liver injuries by a radiologist can facilitate their identification from MR imaging and can guide the most appropriate therapeutic management strategies in oncologic patients.

## Figures and Tables

**Figure 1 diagnostics-12-00867-f001:**
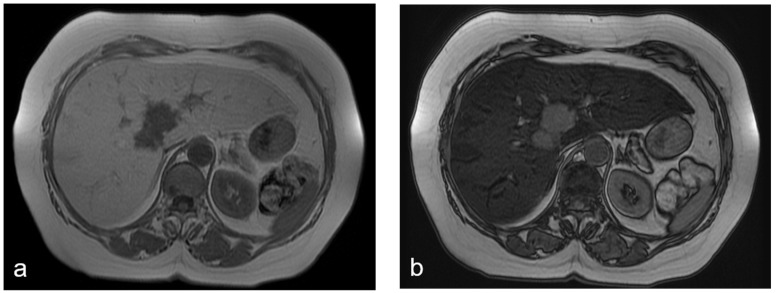
(**a**–**d**) A 68-year-old woman with liver metastases from colon cancer undergoing neo-adjuvant chemotherapy with 5-fluoroacil and irinotecan (FOLFIRI). After chemotherapy, the axial T1-weighted in-phase image (**a**) showed diffuse hyperintensity of the liver with two adjacent hypointense metastatic lesions in segment 4, and the axial T1-weighted opposite-phase image (**b**) demonstrated a loss of signal at the level of the non-tumoral-bearing liver parenchyma, suggesting hepatic steatosis; moreover, the metastatic lesions appeared bright relative to the steatostic liver. About a year after the last chemotherapy and after resection of liver metastases, on MRI, no differences were found in the signal intensity between the axial T1-weighted in-phase image (**c**) and opposite-phase image (**d**) at the level of the liver parenchyma; therefore, we deduced that the steatosis was resolved.

**Figure 2 diagnostics-12-00867-f002:**
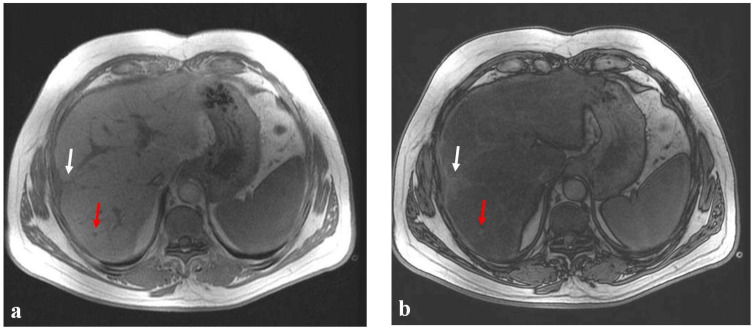
(**a**–**c**) A 38-year-old man with liver metastases from rectal cancer undergoing neo-adjuvant chemotherapy with 5-fluoroacil and irinotecan (FOLFIRI). After chemotherapy, the axial T1-weighted in-phase image (**a**) showed diffuse hyperintensity of the liver with two poorly identifiable metastases, which had good therapeutic responses (white and red arrows), and the axial T1-weighted opposite-phase image (**b**) demonstrated a loss of signal at the level of the non-tumor-bearing liver parenchyma, suggesting hepatic steatosis. The MR imaging proton-density fat fraction (PDFF) map (**c**) showed the percentage of fat calculated from ROIs placed in the two hepatic lobes (right, 26% and left, 21%), in the subcutaneous fat (94%), and in the muscle (3%).

**Figure 3 diagnostics-12-00867-f003:**
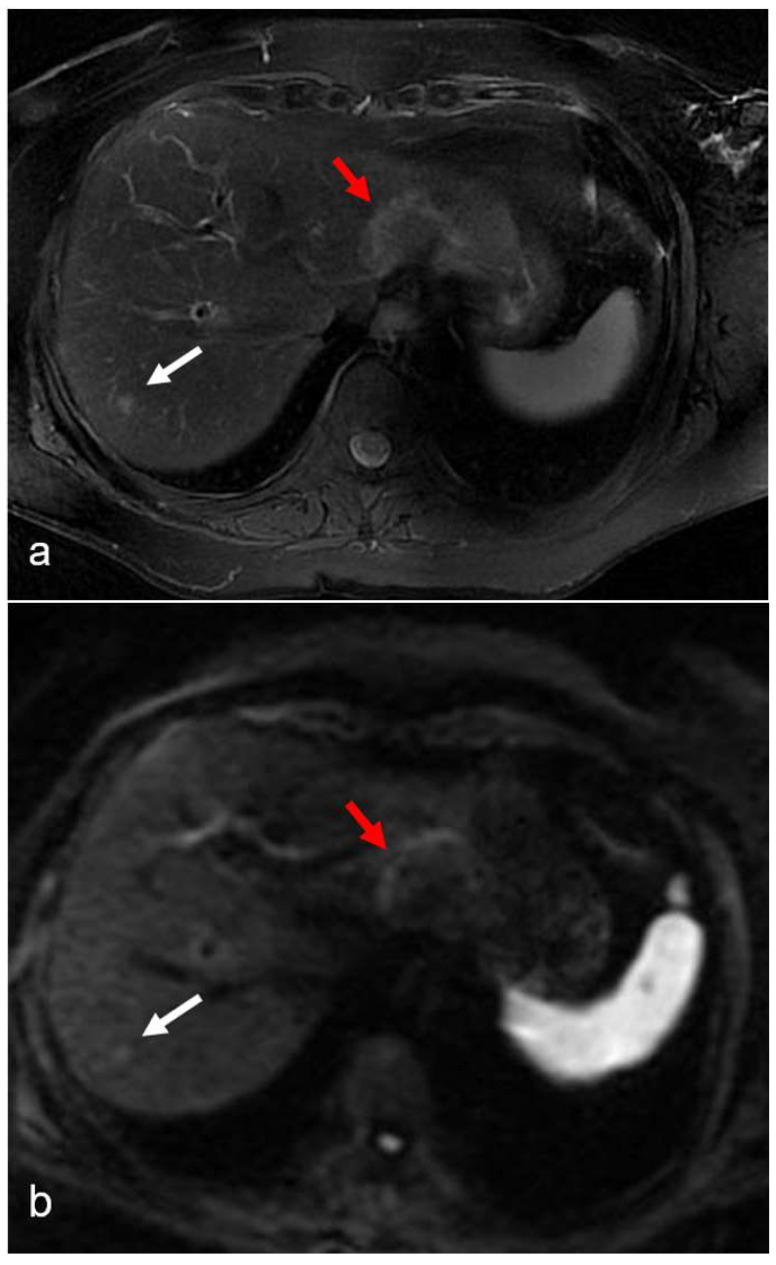
(**a**–**d**) A 55-year-old man with hepatic metastases from rectal cancer undergoing neo-adjuvant chemotherapy with folinic acid, 5-fluorouracil, and oxaliplatin (FOLFOX). After chemotherapy, in the same plane as metastasis of the left lobe (red arrow), a small focal lesion developed in segment 6 (white arrow). The lesion was hyperintense in the T2-weighted image (**a**) and presented a slight restriction in the DW-MRI image, with *b*-values of 1000 s/mm^2^ (**b**). The arterial phase (**c**) of the Gd-EOB-DTPA-enhanced MR image demonstrated homogeneous and strong enhancement at the level of the lesion, which presented a doughnut-like enhancement with hyperintense periphery and a hypointense center in the hepatobiliary phase image (**d**). The findings are consistent with an FNH-like lesion. In the hepatobiliary phase image (**d**), the patchy heterogeneity of the surrounding hepatic parenchyma, with reticular enhancement and scattering of hypodense areas, is also appreciably suggestive of SOS.

**Figure 4 diagnostics-12-00867-f004:**
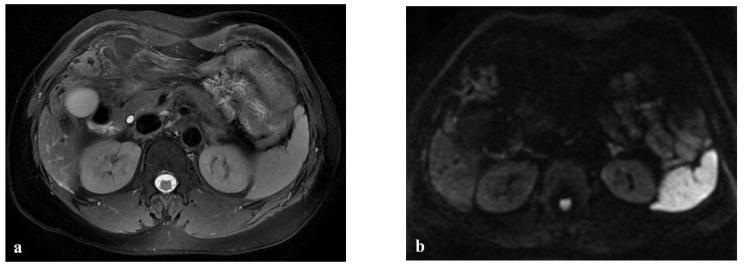
(**a**–**e**) A 62-year-old woman with hepatic metastases from rectal cancer undergoing neo-adjuvant chemotherapy with folinic acid, 5-fluorouracil, and oxaliplatin (FOLFOX). After chemotherapy, a new centimetric lesion appeared in segment 6 (arrow). The lesion was not appreciable on the T2-weighted image (**a**) and on the DW-MRI image, with *b*-values of 1000 s/mm^2^ (**b**) and an ADC value = 1.25 × 10^−3^ mm^2^/s in the reconstructed ADC map (**c**). After the administration of Gd-EOB-DTPA, a lesion with strong central enhancement and hypointense periphery was identified in the arterial phase image (**d**) but was no longer recognizable in the hepatobiliary phase image (**e**). The findings are consistent with a pseudotumor lesion. In the hepatobiliary phase image (**e**), diffuse hepatic reticular hypointensity of the surrounding liver parenchyma is also appreciably suggestive of SOS.

**Figure 5 diagnostics-12-00867-f005:**
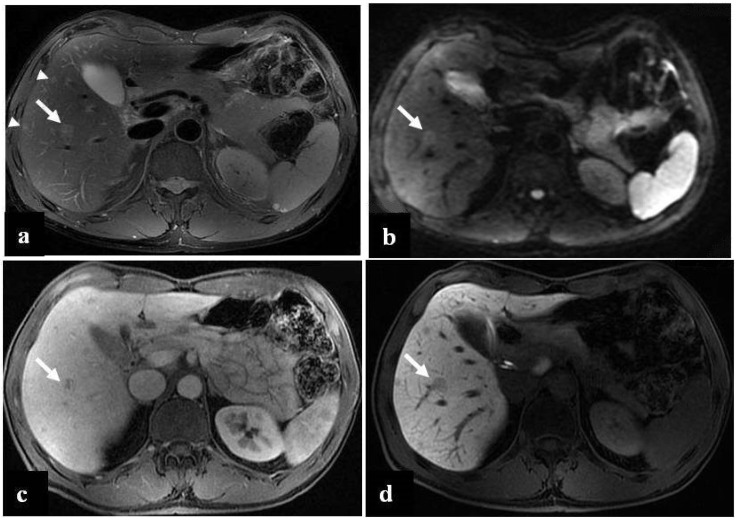
(**a**–**d**) A 49-year-old man with hepatic metastases from colon cancer undergoing neo-adjuvant chemotherapy with folinic acid, 5-fluorouracil, oxaliplatin, and irinotecan (FOLFOXIRI) and with the association of bevacizumab. After chemotherapy, multiple small foci (arrow heads) appeared in the liver parenchyma, with the largest in segment 5 (arrow). This lesion was slightly hyperintense in the T2-weighted image (**a**) and presented slight restrictions in the DW-MRI image, with *b*-values of 1000 s/mm^2^ (**b**). The portal-venous phase (**c**) of the Gd-EOB-DTPA-enhanced MR image demonstrated inhomogeneous enhancement of the lesion, while in the hepatobiliary phase image (**d**), it appeared as a hypointense area. The biopsy of the largest lesion was diagnostic for peliosis.

**Figure 6 diagnostics-12-00867-f006:**
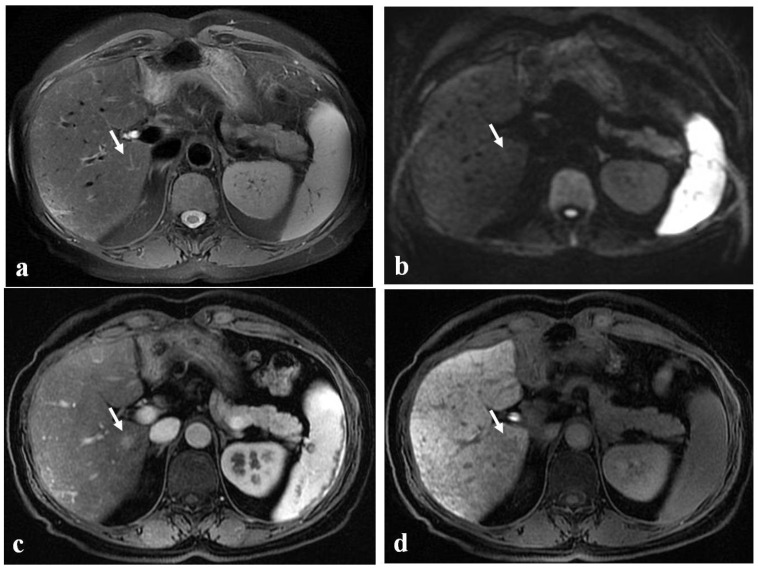
(**a**–**d**) A 52-year-old woman with hepatic metastases from colon cancer undergoing neo-adjuvant chemotherapy with folinic acid, 5-fluorouracil, oxaliplatin, and irinotecan (FOLFOXIRI). After chemotherapy, a new centimetric lesion appeared in segment 7 (arrow). The lesion was not appreciable in the T2-weighted image (**a**) or in the DW-MRI image, with *b*-values of 1000 s/mm^2^ (**b**). After the administration of Gd-EOB-DTPA, the lesion appeared hyperintense in the arterial phase image (**c**) and slightly hypointense in the hepatobiliary phase image (**d**) and was not easily recognizable from the surrounding hepatic parenchyma, which is markedly inhomogeneous and suggestive of SOS. The findings are consistent with an NHR lesion.

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
