# Peer review of "Chemotherapy-Induced Liver Injury in Patients with Colorectal Liver Metastases: Findings from MR Imaging"

_diagnostics, 2022, doi:10.3390/diagnostics12040867_

Round 1

Reviewer 1 Report

Manuscript ID: diagnostics-1594324

Chemotherapy-induced liver injury in patients with colorectal liver metastases: MR imaging findings.

General comment to the authors.

This article handle the topic about the liver pathological state after chemotherapy to the colorectal cancer liver metastases.

In the title of this review, the authors use the word “MR imaging findings”. However, this article have very few MR imaging figures. The reviewer recommend the authors to include the MR images of MRS and PDFF in hepatic steatosis section, and pseudotumor, peliosis, and NRH in SOS section.

Within this review article, the authors include two types of chemotherapy induced liver pathology; fatty liver and sinusoidal obstruction syndrome (SOS). Among the two topics, the description about SOS is redundant, has a lot of duplication, so the reviewer recommend the authors to make it simple. Also, since the SOS part contains several important topics, it should have subheadings to make it easier to understand.

In terms of NRH, the authors described that NRH is the most dangerous form of chemotherapy-induced liver injury. However, ref. #42 says that the presence of NRH has no relationship to the prognosis of SOS. Furthermore, NRH is generally considered to be a hyperplasia of hepatocytes with little pathological significance. So the reviewer recommend the authors to clarify whether the presence of NRH in SOS cases are critical or not and mention whether the NRH in SOS cases are the same pathology or not with the so called NRH observed in normal liver.

Finally, the reviewer think that this manuscript should be checked by a native English speaker since some of the sentences are difficult to understand.

The followings are some of the specific comments to the authors.

Comment 1

Page 1, Introduction section, line 10.

The authors described that “morbidity and mortality to varying degrees”. Please add some reference to this description.

Comment 2

Page 3, line 17

“5.56%” should be “5.7%” (Please standardize the number of digits in the values.)

Comment 3

Page 3, line 27

Please add the vender name and address of “liverMultiScan”, because it seems to be some kind of software or something.

Comment 4

Page 3, Sinusoidal Obstruction Syndrome section, line 9

The authors described that “between 18.1% and 59%”. Please standardize the number of digits in the values.

Comment 5

Page 4, line 28

“oxaplatin-induced SOS” might be “oxaliplatin-induced SOS”

Comment 6

Page 5, line 14

“diffuse-weighted” might be “diffusion-weighted”.

Author Response

Response to Reviewer 1 Comments

General comments response:

  • as requested by the reviewer, four new figures have been added;
  • as suggested, some redundant phrases have been removed in the SOS chapter and a subtitle for the focal chemotherapy-induced hepatopathy has been added;
  • as suggested, NRH significance has been clarified.

Point 1: Page 1, Introduction section, line 10. The authors described that “morbidity and mortality to varying degrees”.

Please add some reference to this description.

Response 1: OK thanks, done

Point 2: Page 3, line 17. “5.56%” should be “5.7%” 

(Please standardize the number of digits in the values.)

Response 2: OK thanks, done

Point 3: Page 3, line 27

Please add the vender name and address of “liverMultiScan”, because it seems to be some kind of software or something.

Response 3: OK thanks, done

Point 4: Page 3, Sinusoidal Obstruction Syndrome section, line 9. The authors described that “between 18.1% and 59%”.

Please standardize the number of digits in the values.

Response 4: OK thanks, done

Point 5: Page 4, line 28 “oxaplatin-induced SOS” might be “oxaliplatin-induced SOS”

Response 5: OK thanks, done

Point 6:Page 5, line 14 “diffuse-weighted” might be “diffusion-weighted”.

Response 6: OK thanks, done

Below I have attached the file of the revised article with the corrections highlighted.

Reviewer 2 Report

  • Page 2 “All of the above limitations of CT are overcome by the use of MRI, that is considered the most sensitive and specific non-invasive technique for detecting, assessing and monitoring steatosis”: CT and its limitations are not mentioned in the text before this sentence, does this come from the reference? Please correct.

  •  

    Page 4: “Patchy heterogeneity of hepatic parenchyma on contrast-enhanced MR images, with a scattering of micronodular or large hypodense or hypointense areas": the term “hypodense” does not belong in a sentence on MR. Please consider removing it
  • Page 5 “MRI findings of hepatic peliosis depend on the stages of blood in these lesions.”: this sentence is repeated
  • Page 5-6 “similar to”, not “similar than”
  • Page 6. NRH is once spelled “NHR”, please check for consistency
  • Page 6 “However, no correlation was demonstrated between NRH and the number of chemotherapy cycles. Nevertheless, short and effective chemotherapy is always recommended to limit NRH.” please clarify: why is a short chemotherapy recommended to limit NRH is no correlation has been observed? And why “effective”?
  • Page 11 – figure 1 legend “Axial T1-weighted in opposing phase”: please consider substituting with”Axial T1-weighted in opposed Phase"
  • Page 11 – figure 1 legend “the steatosis has been resolved”: please change to “the steatosis has resolved”
  • If possible, please add images also for other entities that are mentioned in the paper e.g. pseudotumor, peliosis, cloverleaf/claw-like enhancement.
  • Please consider modifying figure 2 in order also to include the comparison with the behavior of a metastasis in these sequences
  • The reproducibility of MRI in diagnosing SOS has been recently questioned. The Authors should consider including this in their discussion and citing "Magnetic resonance assessment of sinusoidal obstruction syndrome after neoadjuvant chemotherapy for colorectal liver metastases is not reproducible" DOI: 10.1177/0284185120957988

Author Response

Response to Reviewer 2 Comments

General comments response:

  • as requested by the reviewer, four new figures have been added, and figure 2 has been modified;
  • as suggested, the reproducibility of MRI in diagnosing SOS has been added in the discussion and the related article has also been included in the bibliography.

Point 1: Page 2 “All of the above limitations of CT are overcome by the use of MRI, that is considered the most sensitive and specific non-invasive technique for detecting, assessing and monitoring steatosis”: CT and its limitations are not mentioned in the text before this sentence, does this come from the reference? Please correct.

Response 1: OK thanks, done

Point 2: Page 4: “Patchy heterogeneity of hepatic parenchyma on contrast-enhanced MR images, with a scattering of micronodular or large hypodense or hypointense areas": the term “hypodense” does not belong in a sentence on MR. Please consider removing it.

Response 2: OK thanks, done

Point 3: Page 5 “MRI findings of hepatic peliosis depend on the stages of blood in these lesions.”:  this sentence is repeated

Response 3: OK thanks, done

Point 4: Page 5-6 “similar to”, not “similar than”

Response 4: OK thanks, done

Point 5: Page 6. NRH is once spelled “NHR”, please check for consistency.

Response 5: OK thanks, done

Point 6: Page 6 “However, no correlation was demonstrated between NRH and the number of chemotherapy cycles. Nevertheless, short and effective chemotherapy is always recommended to limit NRH.” please clarify: why is a short chemotherapy recommended to limit NRH is no correlation has been observed? And why “effective”?

Response 6: OK thanks, done. This sentence has been removed.

Point 7: Page 11 – figure 1 legend “Axial T1-weighted in opposing phase”: please consider substituting with”Axial T1-weighted in opposed Phase"

Response 7: OK thanks, done.

Point 8: Page 11 – figure 1 legend “the steatosis has been resolved”: please change to “the steatosis has resolved”

Response 8: OK thanks, done. This sentence has been removed.

Below I have attached the file of the revised article with the corrections highlighted.

Round 2

Reviewer 1 Report

The authors revised the manuscript according to the reviewer's comment. The reviewer think the manuscript is OK.